# Characterizing Seasonal Radial Growth Dynamics of Balsam Fir in a Cold Environment Using Continuous Dendrometric Data: A Case Study in a 12-Year Soil Warming Experiment

**DOI:** 10.3390/s22145155

**Published:** 2022-07-09

**Authors:** Shalini Oogathoo, Louis Duchesne, Daniel Houle, Daniel Kneeshaw

**Affiliations:** 1Centre d’Étude de la Forêt, Université du Québec à Montréal, Montreal, QC H3C 3P8, Canada; kneeshaw.daniel@uqam.ca; 2Direction de la Recherche Forestière, Ministère des Forêts, de la Faune et des Parcs du Québec, Quebec City, QC G1P 3W8, Canada; louis.duchesne@mffp.gouv.qc.ca; 3Science and Technology Branch, Environment Canada and Climate Change, Montreal, QC H2Y 2E7, Canada; daniel.houle@ec.gc.ca

**Keywords:** tree growth, point dendrometer, seasonality, phenology, treenetproc, boreal forest

## Abstract

Historical temperature records reveal that the boreal forest has been subjected to a significant lengthening of the thermal growing season since the middle of the last century, and climate models predict that this lengthening will continue in the future. Nevertheless, the potential phenological response of trees to changes in growing season length remains relatively undocumented, particularly for evergreen boreal tree species growing in cold environments. Here, we used the recently defined zero growth (ZG) concept to extract and characterize the metrics of seasonal radial growth dynamics for 12 balsam fir trees subjected to a 12-year soil warming experiment using high resolution radius dendrometer measurements. The ZG concept provides an accurate determination of growth seasonality (onset, cessation, duration, growth rates, and total growth) for these slow-growing trees characterized by significant shrinkage in tree diameter due to dehydration in the winter. Our analysis revealed that, on average, growth onset starts at day 152 ± 7 (±1 SE, 31 May–1 June) and ceases at day 244 ± 27 (31 August–1 September), for a growing season duration of about 3 months (93 ± 26 days) over a 12-year period. Growing season duration is mainly determined by growth cessation, while growth onset varies little between years. A large part (80%) of the total growth occurs in the first 50 days of the growing season. Given the dynamics of growth, early growth cessation (shorter growing season) results in a higher average seasonal growth rate, meaning that longer growing seasons are not necessarily associated with greater tree growth. Soil warming induces earlier growth cessation, but increases the mean tree growth rate by 18.1% and the total annual growth by 9.1%, on average, as compared to the control trees. Our results suggest that a higher soil temperature for warmed trees contributes to providing better growth conditions and higher growth rates in the early growing season, when the soil temperature is low and the soil water content is elevated because of snowmelt. Attaining a critical soil temperature earlier, coupled with lower soil water content, may have contributed to the earlier growth cessation and shorter growing season of warmed trees.

## 1. Introduction

The examination of historical daily temperature records reveals that the Canadian forest has been subjected to significant climate warming since the middle of the last century, with even greater warming in the boreal forest, typically characterized by colder temperatures [1]. In addition to the general warming trend, an average lengthening, by about 15 days, of the thermal growing season (days with daily mean temperature above 5 °C), starting 6 days earlier and ending 9 days later, coupled with increased cumulative growing degree days, has been reported for the 1948–2016 period [1]. These trends in thermal growing season length are predicted to continue throughout this century, potentially adding more than a month of favorable conditions for the yearly growth of trees and plants [2,3]. Together with the fertilization effect of rising atmospheric CO_2_ and N inputs on tree growth, the lengthening of the growing season could potentially contribute to a large increase in C sinks observed over the last decades in northern temperate and boreal forests [4,5].

Nevertheless, the potential phenological response of trees to changes in climate seasonality remains relatively undocumented, particularly for the cold environment of the boreal forest. Several approaches have been used to document the seasonal growth dynamics of trees, each with their strengths and weaknesses. Phenological metrics linked to vegetation activity onset, cessation, duration, and peak can be derived over a large spatial scale from satellite observations of the photosynthetic activity of vegetation [6,7]. These metrics are, however, generally restricted to the analyses of pixels with a strong detectable annual cycle of vegetation, filtering out evergreen boreal coniferous forest regions [8,9]. Characterizing vegetation dynamics from satellite information in cold environments characterized by a short growing season, short daily sunshine duration in winter, persistence of snowpack in early spring, and the dominance of coniferous tree species remains challenging [10]. Moreover, phenological metrics derived from satellite images do not inform about species-specific variations, nor seasonal tree growth dynamics and tree productivity. Indeed, the greening effect documented from the satellite-derived normalized difference vegetation index (NDVI) does not always translate into ground-observed phenological trends or tree growth rates [7,11,12]. The extraction of metrics of tree seasonality using remote sensing techniques also involves technical challenges for taking into account the quality of the sensors and other distortions in the satellite images, and large discrepancies exist between ground-observed and satellite phenological transitional dates [8]. Phenological metrics derived from satellite information are also restricted in terms of temporal and spatial resolution, which are, for example, 16 days and 250 m resolution for MODIS NDVI images [13].

Transitional phenological dates can also be assessed from ground-based observations, including networks of leaf and bloom measurements of representative cloned cultivars (e.g., [14]), networks of digital phenological cameras (e.g., [15]), and local monitoring initiatives [16]. However, the assessment of phenological metrics, such as leafing and blooming, are also generally limited to deciduous trees, and they do not inform on tree growth dynamics over the entire growing season and on resultant tree productivity, particularly for the evergreen boreal tree species.

At the tree level, cambial activity and xylem formation (xylogenesis) of boreal species can be characterized using cellular analysis of repeatedly sampled micro-cores of newly formed xylem throughout the growing season [17] and by the pinning method [18]. These techniques are, however, laborious and time consuming in addition to damaging to the trees, limiting the long-term monitoring of individual trees. Characterization of growth dynamics using these techniques is also limited in terms of temporal resolution by the sampling (or pinning) interval. In addition, shifting sample points along the stem during the growing season generates additional variability in the measurements.

Over the past decade, a growing number of studies have used electronic dendrometers to track variations in tree radius or circumference (radius hereafter) at high temporal resolution in order to document the critical dates of xylem phenology [19,20,21,22,23,24,25]. These metrics of tree growth phenology include the timing of growth onset, cessation, and maximum growth rate, along with growing season duration and cumulative seasonal growth.

However, there are also limitations to determining phenological transition dates and quantifying tree growth using point dendrometers. In dendrometer data, both the signal of irreversible tree growth and the reversible signal of stem size, due to water storage depletion and replenishment of wood tissues, are confounded. These variations correspond to three distinct previously documented phases (growth increment, contraction or shrinkage, and expansion or recovery) of the circadian cycle of stem radius variation [26,27,28] and may be extracted with the help of a function for stem cycles definition included in the Dendrometer R package [29]. In addition to the diurnal cycle that characterizes stem size variation, periods of stem shrinkage may occur over a longer period, from a few days during the growing season, due to dry conditions of air and soil [30,31], to seasonal (i.e., months) during the winter, due to the dehydration of wood tissues [22,32]. During these periods of stem shrinkage, the signal is still characterized by cycles of stem contraction and expansion, but at a value below a precedent maximum stem size. Zweifel et al. [30] thus defined the zero growth (ZG) concept, according to which no growth occurs during periods of stem shrinkage. Using this concept, the signal of stem size variation may be partitioned into a growth (GRO) phase, which begins when the previous maximum stem radius is exceeded and ends when stem shrinkage occurs, and a tree water deficit (TWD) phase, which begins when the stem size decreases below a precedent maximum. The GRO phase derived from the ZG concept is slightly different from the increment phase of the circadian cycle approach [28]. The beginning of the GRO phase is determined from a previous maximum stem size, while the increment phase begins when the stem size exceeds the morning maximum.

One advantage of the ZG approach is that the GRO signal can be cumulated over the growing season to estimate yearly cumulative growth, which is not possible with the three-cycle approach. In the three-cycle approach, the same variation in stem size may be considered to be an increment phase when it occurs several days in a row; this can thus greatly overestimate total growth, if the stem radius variation during the increment phases is cumulated over the whole growing season [24,28]. Nevertheless, it should be noted that both approaches can be reconciled with a slight change in definition. Indeed, the TWD signal derived from the ZG concept could be decomposed into phases of stem contraction, shrinkage or dehydration, and stem expansion, recovery, or rehydration, thus restricting the increment phase to the GRO phase.

Another advantage of the ZG concept is that the timing of seasonal growth onset and cessation can be respectively derived from the maximum stem radius recorded over the previous season and during the current growing season [22]. From these metrics, growing season duration, cumulative growth, and average growth rate can also be derived. This simplification is particularly relevant for slow-growing trees inhabiting cold environments characterized by a long period of winter dehydration for which spring growth onset may be confused with stem rehydration before the beginning of cambial growth [32,33]. In this context, the three-cycle approach does not seem appropriate for precisely identify the timing of growth onset and cessation [24], and deriving the timing of growth onset during the spring rehydration phase is practically impossible without resorting to the ZG concept, or without the use of a complementary technique, such as micro-coring [27,32,33]. Seasonal growth dynamics have already been estimated from raw dendrometer data by fitting a statistical model to the data [19,20,34,35,36]. As opposed to the ZG concept, this approach results in a continuous growth curve (CG approach) characterizing the seasonal growth dynamics. Even with this approach, it remains difficult to precisely determine growth onset and cessation, and the beginning of the growing season remains practically undistinguishable from the early spring rehydration period.

The aims of this study are to test the use of the ZG concept for extracting and characterizing the metrics of seasonal radial balsam fir growth dynamics using high temporal resolution dendrometer measurements and to assess interannual variability, as well as the effects of a 12-year soil warming experiment on radial growth. In addition to potentially promoting better growth throughout the growing season, the early warming of soil temperatures in spring can induce favorable conditions for xylogenesis when air temperatures are above critical temperatures [37], but delayed soil warming due to snow cover still prevents tree growth [38,39]. We thus hypothesized an earlier growth onset, a longer duration of the growing season, and a higher annual radial growth for warmed-soil trees as compared to controls.

## 2. Methods

### 2.1. Study Site

The experimental area is located at the boundary of the Lac Laflamme watershed (47°19′31.36″ N; 71°07′43.15″ W, 740–760 m.a.s.l.), which is one of the three study watersheds that are part of the Québec Forest Ecosystem Research and Monitoring Network (Réseau d’Étude et de Surveillance des Écosystèmes Forestiers; RESEF). The forest is dominated by mature balsam firs (*Abies balsamea* (L.) Mill.), with a small component of white spruce (*Picea glauca* (Moench) Voss) and paper birch (*Betula papyrifera* Marsh.). The site is on Precambrian charnockitic gneiss covered by sandy till, with a terrain slope of 15%. Climate norms (1981–2010) were −0.18 °C and 1460 mm for mean annual air temperature and mean annual precipitation, respectively. The forest is located in the Laurentian Mountains, Québec, Canada, and underwent defoliation by the hemlock looper (*Lambdina fiscellaria*, Guenée) from 2012–2014 [40]. The 2012 defoliation was concurrent with a severe drought in the area in July 2012. With only 23 mm of rain in July as compared to the 143 mm long-term average, it was the driest summer month ever observed since 1960 [41].

### 2.2. Soil Warming Experiment

The soil warming experiment was set up in autumn 2008 and the short-term (2–3 years) effects on soil ion fluxes and mineralizable carbon, as well as on xylogenesis and tree growth of mature balsam firs, have been documented [34,42,43,44]. The experiment consists of 12 balsam firs (6 heated and 6 control) around which 70 m of heating or non-heating cables were buried in a spiral pattern (0.9 to 2.5 m radius, approximately 30 cm between successive turns) at the interface of the organic and mineral soil (5–10 cm belowground). The soil temperature below each tree was measured continuously with a thermistor (model 107-L, Campbell Scientific, Logan, UT, USA) positioned between the cables and linked to a data logger (model CR1000, Campbell Scientific, Logan, UT, USA). The continuous measurements were averaged and recorded at 15 min intervals. The warming cables of the 6 heated trees were activated and deactivated with a data logger when the average difference of temperature between the heated and non-heated soils reached 3 °C and 5 °C, respectively, with the objective of maintaining an average difference of 4 °C. Since 2009, the soils were heated from the beginning of the spring snow melt until the first frost days in the fall.

Each of the twelve trees in the experiment was equipped with radius dendrometers (model DR1, Ecomatik, Munich, Germany), south oriented, and positioned at approximately 2 m height to ensure being above the snow cover. Stem radial variation was measured continuously, and data were averaged and recorded every 15 min using the same data logger used for the soil measurements.

The soil warming treatment was initially nested within two levels of artificial precipitation (no added NH_4_NO_3_ and three times the natural NH_4_NO_3_ concentration in ambient precipitation) applied to the canopy every week from mid-June to mid-September from 2009 to 2011 using a system of reservoirs, pumps, valves, pipes, and sprinklers. No short-term effects of this N-enriched precipitation on tree growth were detected [34], so we did not consider this treatment in our analysis.

### 2.3. Characterization of Tree Growth Seasonality from Dendrometer Data

The extraction of parameters defining the seasonal dynamics of tree radial growth was performed with the help of the *treenetproc* package in R, V 0.1.4 [22,45,46]. These parameters included growth onset and cessation (day of year; DOY), growing season length (days), total annual growth (µm), and the average daily growth rate (µm/day). Spring growth onset is defined as the DOY at which the stem radius surpasses the maximum value recorded in the past year, and growth cessation is considered as the DOY at which the maximum value is reached (Figure 1, [22]). The length of the growing season is the difference between these two values, and total annual growth is the difference between the radius measurements corresponding to the beginning and end of growth. Finally, the average daily growth rate is the total annual growth divided by the length of the growing season.

Prior to the seasonal tree growth parameter extraction, the raw 15 min dendrometer data were visually examined and validated for any erroneous data that could be manually removed. As input, the *treenetproc* R package requires raw dendrometer data and air temperature. The purpose of the latter is to identify frost shrinkage periods in order to avoid assigning dendrometer data as outliers [22]. Internal data logger temperature data were used as an indicator of air temperature. This ensures accurate synchronicity between the ambient temperature and dendrometer measurements. The *treenetproc* package was then used to temporally align the dataset, filling the dataset via linear interpolation for small gaps (fewer than 3 time steps), and correcting for erroneous shifts and jumps in the manually cleaned dendrometer data [22]. We adjusted the parameters controlling the rigidity of outlier (*tol_out*) and jump (*tol_jump*) detection so that only jumps in the data associated with dendrometer adjustment or replacement were applied.

The *treenetproc* R package, used to determine growth onset and cessation (*grow_seas*), includes a tolerance factor parameter to adjust the determination of growth onset and cessation (*tol_seas*). This factor allows us to take into account the asymptotic shape of the seasonal growth pattern at the beginning and end of the growing season, as well as the presence of erratic measurements during these periods that can influence the determination of the beginning and end of growth [22]. The default value is set to 5%, so that the start and end of the growing season corresponds to the DOY when 5% and 95% of the total annual growth is completed, respectively. We set the parameters to the default value for the main analysis and also set the timing to correspond to the time when 80% and 90% of the total annual growth was completed (*tol_seas* value of 10% and 20%). We varied the *tol_seas* parameter only for the determination of growth cessation because this parameter has little influence on the timing of growth initiation, which is generally characterized by a much steeper slope of stem radius variation compared to the growth cessation phase.

The 12 dendrometers were operational from March 2009 to December 2021. Since the calculation of growth onset depends on the maximum radius measurements of the previous year, seasonal parameters were characterized for each tree, starting in the second year of measurements (2010) until 2021.

### 2.4. Statistical Analyses

Interannual variability in parameters characterizing tree growth seasonality (growth onset and cessation, growing season length, total annual growth, and the average daily growth rate), along with the soil warming treatment effect, were assessed by means comparisons and from mixed between-subjects (control/heated) and within-subjects (years) analysis of variance of the *trimmed mean* [47]. The *trimmed mean* excludes the lowest and the largest values of the distribution and computes the arithmetic mean on the remaining values. Analysis of variance based on *trimmed means* has been proposed as a robust alternative to classical statistical methods, when distributional assumptions are not met [47]. The *bwtrim* function of the *WRS2* R package V 1.1-1 was used for this purpose [47]. We performed one analysis considering the 12 growing seasons (2010–2021) of the 7 trees that survived the 2012 event and another by including the 3 growing seasons (2010–2012) of the 5 trees that did not survive. Additionally, the strength of the monotonic association among the parameters characterizing the seasonal dynamics of radial growth of individual trees (i.e., experimental units) were analyzed using Spearman’s rank-order correlation.

## 3. Results

### 3.1. Soil Temperature and Dendrometer Data

The snow cover melted earlier, spring soil warming occurred, on average, 25 days earlier (heated and control soils reach 2 °C on day 113 and 138, respectively, Figure 2), and the mean annual May to October temperature of the heated soils was, on average, 3.8 ± 0.2 °C (±SE) higher compared to that of the unheated soils over the period 2009–2021.

Based on the dendrometer data, the seasonal parameters could not be evaluated for some trees in some years. The defoliation and severe drought that occurred in the area in July 2012 had a marked impact on the growth of trees in the subsequent years, and 5 trees died in 2013 from these events (trees 102, 111, 115, 124, and 125). The stem of another tree failed during an episode of sustained high winds that occurred from 12 to 15 August 2021 (tree 122). The parameters used to define the growing season could not be evaluated for three other seasons due to malfunctioning dendrometers. For two of these seasons (tree 111 in 2011 and tree 124 in 2012), the dendrometers had to be replaced due to water infiltration and ice formation in the sensor that prevented linear movement of the potentiometer. This phenomenon was observed in one case at the end of March and in the other during the month of November, two periods characterized by alternating rain and snow. Following this observation, plastic bibs were stapled to the trunk of the tree above the sensor to protect it from rain and snow, which prevented the problem from recurring. Finally, the 2018 season could not be evaluated for one tree (122) because the potentiometer had reached its maximum stroke during the growing season, and the late readjustment of the sensor made it impossible to join the series of measurements before and after the period with missing measurements. In the end, the studied database includes 4.83 million soil temperature measurements and 4.85 million stem radius measurements at a temporal resolution of 15 min, from which we characterized the tree growth seasonality of 94 of the 144 seasons (Appendix A).

### 3.2. Tree Growth Seasonality

Annual, treatment, and overall averages of parameters characterizing the seasonal dynamics of balsam fir radial growth are presented in Table 1. Over the 12 studied years, the growth onset of balsam fir trees occurred, on average, on day 152 ± 7 (31 May–1 June; Table 1 and Appendix A). The earliest seasonal onset was observed in 2013, 10 days earlier than the average (day 142, 22 May), while the latest season, observed in 2019, started 11 days later than the average (day 163, 12 June). The average growth onset of control (DOY 151) and heated (DOY 152) trees did not differ significantly, and the soil warming treatment did not influence yearly variability (*P* ≥ 0.1 for treatment–year interaction; Table 2).

Growth cessation (95% of yearly growth completed) occurred, on average, on day 244 ± 27 (31 August–1 September). The earliest end date was observed in 2021, 33 days sooner than the average (day 211, 30 July), while the latest end date was observed in 2018, when the season ceased 26 days later than the average (day 270, 27 September). The growth cessation of warmed trees (day 240) occurred, on average, 1 week (7 days) earlier than for the control trees (day 248, *P* < 0.1; Table 1, Table 2 and Appendix A), and the effect of the soil warming treatment varied among years (*P* < 0.1 for treatment–year interaction; Table 2), with maximum differences of 24 and 60 days for the years 2017 and 2018, respectively.

On average, the resultant growing season length was evaluated as 93 ± 26 days, with longer growing seasons for control trees (96 days) than for heated trees (88 days, *P* < 0.001; Table 1, Table 2 and Appendix A) and a less significant interaction of the soil warming treatment with the years (*P* < 0.1, Table 2). On average, the total annual growth ranged from 0.66 mm in 2014 to 1.58 mm in 2017 and averaged 1.16 mm over the 12-year period (Table 1, Table 2 and Appendix A). The total annual growth of the heated trees is 9.1% higher than for the control trees (*P* < 0.01; Table 1 and Table 2), and this difference does not vary significantly over the years (*P* > 0.1 for treatment–year interaction, Table 2). Trees grew, on average, at a mean daily growth rate of 13.7 µm per day, with an 18.1% higher growth rate for heated (15.0 µm d^−1^) compared to control (12.7 µm d^−1^) trees (*P* < 0.001, Table 1, Table 2 and Appendix A). There is no significant difference in the effect of the warming treatment on tree growth rates within years (*P* > 0.1 for the treatment–year interaction, Table 2). As for total annual growth, the lowest overall average growth rates were observed in 2013 (8.0 µm d^−1^) and 2014 (5.9 µm d^−1^).

While, on average, 95% of the total annual growth is completed by day 244 (Table 1), 80% of total growth is completed 43 days earlier (DOY 201, Appendix A), and 90% of the total growth is completed 25 days earlier (DOY 222, Appendix A). Considering the lower threshold value for characterizing the end of the growing season (80% or 90% of the total annual growth instead of 95%) considerably reduces the between-trees and within-years variability of this seasonal parameter and the corresponding growing season length (Table 1 and Appendix A). The effect of the warming treatment on the timing of growth cessation and growing season is more tenuous when considering a lower threshold for the characterization of growth cessation (Table 2). Differences in growth cessation, growing season length (95% of yearly growth completed), and daily growth rates between heated and control trees remain statistically, but marginally, significant (*P* < 0.1) when including the 3 growing seasons of the 5 trees that did not survive the combined defoliation and drought event of 2012 (Appendix A). Nevertheless, no difference in the total annual growth between treatments was detected.

### 3.3. Correlations among the Seasonal Parameters

The analysis of the correlation among parameters characterizing radial growth seasonality revealed that the total annual growth is closely and positively related to the average daily growth rate (*p* = 0.88, *P* < 0.001), the latter being negatively related to the timing of growth cessation (95% of yearly growth completed, *p* = −0.59, *P* < 0.001) and to growth duration (*p* = −0.60, *P* < 0.001), but not to the timing of growth onset (*p* = −0.04, Table 3). The duration of the growing season is closely and positively related to the timing of growth cessation if a 95% threshold was considered (*p* = 0.93, *P* < 0.001), and the relationship remain statistically significant for the two other thresholds analyzed (80% and 90%). Finally, the timing of growth cessation is positively related to the timing of growth onset, with a tighter relationship when the 80% threshold was considered (*P* < 0.001, Table 3).

## 4. Discussion

### 4.1. Tree Growth Seasonality

This study is the first to document seasonal radial growth dynamics of trees in cold environments over a relatively long period from high resolution radius dendrometer measurements using the ZG concept. Our analysis over the 12-year period revealed that the onset of growth of balsam fir trees occurs, on average, on day 152 ± 7 (±1 SE, 31 May–1 June). The timing of growth onset and cessation of naturally growing balsam fir trees located at a nearby site (approximately 500 m away) has already been estimated from raw dendrometer data using the CG approach [20]. From this analysis, an earlier average growth onset (by 12 days) was identified for the 2004–2010 period. For the 2010 and 2011 growing season, D’Orangeville et al. [34], also using the CG approach, identified that growth onset occurred 25 and 28 days sooner, respectively, for the same trees studied here.

The earlier onset of growth derived from the CG approach may be induced by the confusion between spring growth onset and stem rehydration occurring before the beginning of cambial growth (Figure 1). The ZG concept by which growth onset was determined from the previous maximum stem radius recorded over the previous season apparently makes it possible to accurately identify the beginning of cambial growth, even during the phase of tree stem enlargement due to the rehydration of trees in early spring (Figure 1). A close look at Turcotte et al.’s study [32] revealed that the onset of wood formation in black spruce trees, as determined by cellular analysis of repetitively sampled micro-cores, coincides with the time when the radius of the tree exceeds the maximum value recorded by dendrometers over the previous season. This growth onset was May 27 in 2005, which is very close to our average growth onset estimate over the 12-year period (5 days earlier, DOY 147 vs. 152). Turcotte et al.’s study [32] is among the few documenting the large magnitude of tree shrinkage occurring during the winter period for trees growing in the cold area of the Canadian boreal forest. In most cases, trees are monitored during the frost-free period, but according to the ZG concept, trees should be monitored continuously for growth onset determination, including during the late fall and winter period of stem shrinkage due to stem dehydration, and during the early-spring stem expansion period due to stem rehydration [22].

Our dating of seasonal growth cessation shows much greater interindividual and interannual variability than growth onset estimates, as well as greater divergence from growth cessation dated using the CG approach (Table 1; [20,34]). This is consistent with earlier observations that this metric shows greater variability as compared to growth onset, which was not as well explained by the climatic variables over the 2004–2010 period for trees growing nearby [20]. The asymptotic curve of the late growing season growth makes the determination of growth cessation more complex when compared to the early growing season period, which is characterized by more abrupt changes in the dendrometer records (Figure 1 and Appendix A; [20]). The CG approach is very sensitive to stem shrinkage due to dry conditions occurring at the end of the growing season, or to the onset of winter desiccation in late fall, as the asymptote of the modeled progressive growth curve will be underestimated by attempting to minimize the sum of the squared residuals at the end of the growing season [20]. Dating of growth cessation using the ZG approach is also a challenge. From this concept, identification of growth cessation relies on the maximum radius size recorded in a given year. Because this maximum value depends on a single high-resolution observation, very small erratic fluctuations in the signal occurring at the end of the growing season can lead to great variability in determining the end-of-season dates, given the asymptotic shape of the dendrometer data during this period. To overcome this problem, Knüsel et al. [22] proposed the use of a tolerance factor for characterizing growth onset and cessation by excluding the first and last observations corresponding to a certain proportion of the total annual growth. A sensitivity analysis of this factor for the application of the ZG concept revealed a much greater influence on dating of growth cessation than growth onset, for the reason mentioned above. The use of the lower threshold value reduces the variability in the timing of growth cessation by avoiding the slow growth phase of the late growing season, while capturing a large part of the annual growth (Appendix A). The time it takes to reach a certain threshold (e.g., 50% of yearly growth completed) and to move from one threshold to another could also be studied [36].

Correlation analyses revealed that the length of the growing season was closely related to the timing of growth cessation (*p* = 0.93, *P* < 0.001, Table 3), but not to the timing of growth onset. This indicates that the late and slow-growing part of the season is the most determinant of the seasonal growth duration. However, growth cessation and duration are only marginally significantly related to total annual growth (*p* = −0.21 to −0.22, *P* < 0.1), indicating that growth phenology metrics (onset, cessation, duration) are not the only drivers of tree growth and productivity. Seasonal growth was also a result of the growth rate during the growing season, with the latter being strongly positively related to cumulative seasonal growth (*p* = 0.88, *P* < 0.001, Table 3). The average growth rate over the season is also negatively related to the timing of growth cessation (*p* = −0.59, *P* < 0.001). Given the dynamics of cell growth, a season that ends abruptly will result in a higher growth rate over a short period of time, whereas seasons that last longer will have a protracted period of lower average growth and a lower overall growth rate. The weak relationship observed with the start of the growing season underscores the importance of late-season weather conditions in determining growing season duration. Using the CG approach, Duchesne et al. [20] similarly observed that growing season duration for balsam fir trees growing in the vicinity of the present study from 2004 to 2010 was primarily dictated by growth cessation, while growth onset was much less variable between years. The synchronicity of growth onset over the years suggests that day length or photoperiod, a parameter that is much more constant from year to year compared to spring climatic conditions, may be an important factor determining the onset of the growing season [48,49]. In the context of climate change, this raises concerns about the adaptation of tree species to the future synchrony of climate seasonality and photoperiod.

### 4.2. Soil Warming Effect

We hypothesized earlier growth onset, longer duration of the growing season, and superior annual radial growth for warmed-soil trees compared to controls. Contrary to our hypothesis, our results show that growth onset of control and heated trees did not differ significantly, relativizing the potential effects of earlier snowpack melting and higher soil temperature on boreal tree seasonal growth onset. The latter is apparently more closely linked to photosynthetically active radiation and air temperature [20]. Moreover, contrary to our hypothesis, the growth of warmed trees ceased, on average, 1 week earlier and lasted 8 days less than that of the control trees. Finally, a part of our hypothesis was verified, given that (despite a shorter growing season) warmed-soil trees had an 18.1% higher average growth rate and 9.1% higher total annual growth. This suggests a positive effect of soil heating early in the growing season, but a negative effect later in the growing season. Among the factors potentially favoring a higher growth rate, it has been shown that soil warming increased the cumulative nutrient fluxes (measured using ion-exchange membranes) in the forest floor and the mineral horizon during the first 3 years of this experiment [43]. A higher soil temperature for warmed trees (Figure 2) can also contribute to providing better growth conditions and higher growth rates, particularly in the early growing season, when soil water content is elevated because of snowmelt [50]. The effect of soil warming in the late growing season may, however, exacerbate the general declining trend in soil water content throughout the growing season until September, when higher precipitation and lower plant water uptake contributed to the soil water replenishment at values similar to those noted in the spring [50]. In 2006, an anomalous early growth cessation (DOY 187, 6 July) and a low seasonal radial growth (550 µm) for trees growing nearby were related to an early peak in soil temperature [20,51]. No other climatic variable was found to be correlated with the timing of growth cessation over the 2004–2010 period. Therefore, exceeding a critical soil temperature (Figure 2) early in the season, coupled with lower soil water content, may have contributed to the earlier growth cessation and shorter growing season of warmed trees. Another likely explanation for the early growth cessation of fast-growing trees is the control of photosynthesis by sink activity [52]. According to this concept, higher seasonal productivity leads to earlier growth cessation due to carbon sink limitations. However, this concept is apparently not supported by consistent observations from the experiments [53].

As reported after a 3-year-long manipulation with soil warming at the studied site [42], a twin experiment at two other sites showed no significant effect of soil warming on the phenological phases of cell enlargement, wall thickening, and lignification, nor on the number of cells produced by black spruce trees (*Picea mariana,* Mill. BSP) after 3 [54], and 6-year-long manipulations [55]. These studies characterized the xylogenesis of trees from cellular analysis of weekly sample micro-cores. The temporal resolution of the method used makes it difficult to detect a difference of a few days in tree growth phenology in response to the soil warming treatment, as we document in the present study. Here, the relatively small treatment effect could be detected due to the high temporal resolution of the approach used to document tree growth phenology. Treatment effects may also be greater over a 12-year period compared to analyses conducted over a 3- or 6-year period. Consistent with these studies, our analysis nevertheless reveals a relatively small effect of the soil warming treatment compared to the interannual variability in tree growth seasonality induced by other climatic parameters, notably air temperature [42,54,55].

### 4.3. Effects of Drought and Insect Defoliation

A total of 5 of the 12 instrumented trees died as a result of insect defoliation and the concurrent drought that occurred in 2012. The results indicate that 4 of the 5 dead trees showed the lowest radial growth among the 12 trees in the years preceding the event (Appendix A), suggesting that the least vigorous trees are primarily those that succumbed. The impact on the growth of the surviving trees is noticeable 2 to 4 years after the event, when growth was significantly reduced compared to other years (Appendix A). The death of these trees is unfortunate for the monitoring of the ongoing experiment, but these trees continue to be monitored with dendrometers. This monitoring represents an opportunity to document the etiology of tree death. Diurnal cycles are observable in the data, even several years after tree death. Analysis of these cycles, coupled with information on sap flow in living trees, could also unravel the influence of ambient air moisture and sap flow on stem radial variations.

## 5. Conclusions

This study is the first to document seasonal radial growth dynamics of trees in cold environments over a relatively long period of 12 years using the ZG concept. The latter seems well suited to determine the metrics of growth seasonality from dendrometer data for slow-growing trees subjected to significant winter shrinkage in tree diameter due to dehydration. The results contribute to our knowledge on tree growth seasonality in the boreal forest by showing that growth occurs over a short period of about 3 months, with most of the growth being achieved in the first 50 days of the growing season. The timing of growth cessation is the stronger determinant of growing season duration, while growth onset has a smaller influence. The average growth rate of trees is higher during growing seasons that end early and is lower as seasons lengthen, meaning that longer seasons are not necessarily the most productive. Soil warming induced earlier growth cessation and shortened the growing season, but increased the mean growth rate and total annual growth of the trees. These results provide crucial information on the potential phenological response of trees to changes in thermal growing season length to better anticipate the effects of climate change on boreal forest productivity. However, as is often the case in this type of study, the results are based on the monitoring of one tree species at a single site. Additional studies are needed to improve our knowledge of the phenological response of boreal forest tree species to climate change.

## Figures and Tables

**Figure 1 sensors-22-05155-f001:**
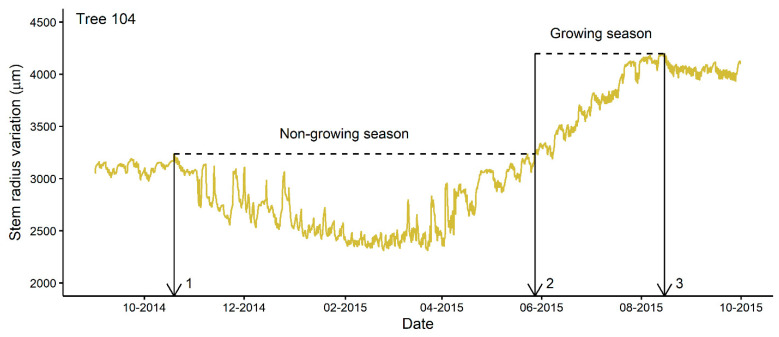
Example of characterization of a growing season using stem radius variations of a balsam fir tree recorded by a dendrometer at a temporal resolution of 15 min. The day of the year corresponding to the maximum value recorded the previous season (1), the day of the year at which stem radius surpasses the maximum value recorded in the past year (2), and the day of the year at which the maximum value is reached (3) are labelled.

**Figure 2 sensors-22-05155-f002:**
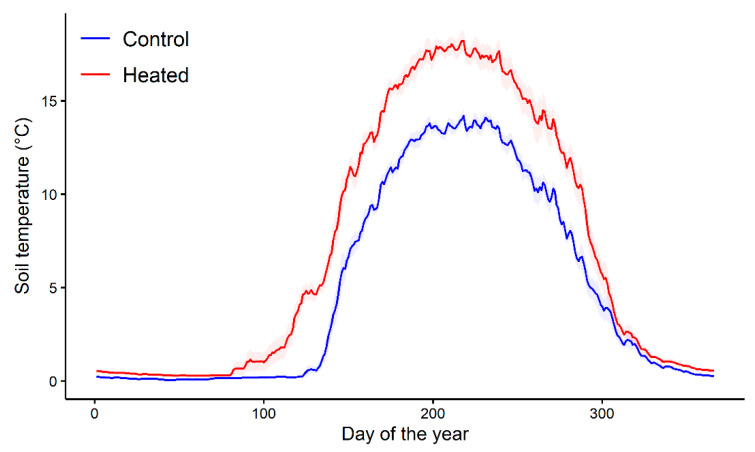
Average annual pattern of daily mean soil temperature (2010–2021) of control and heated plots. The shaded area represents the standard error.

**Table 1 sensors-22-05155-t001:** Average metrics of the seasonal radial balsam fir growth dynamics for control and soil-heated trees over a 12-year period (2010–2021).

	Years												Means
	2010	2011	2012	2013	2014	2015	2016	2017	2018	2019	2020	2021	
**Growth onset (DOY; 5% of yearly growth completed)**													
**Control**	147	155	147	140	149	149	155	156	157	164	157	146	151
**Heated**	145	160	148	146	157	148	153	153	154	162	155	145	152
**Overall**	146	157	147	142	152	149	154	155	156	163	156	146	152
**Growth cessation (DOY; 95% of yearly growth completed)**													
**Control**	249	232	236	241	273	241	264	248	294	259	242	214	248
**Heated**	254	232	242	242	261	226	268	223	234	242	232	206	240
**Overall**	252	232	239	242	268	235	266	237	270	252	238	211	244
**Growing season duration (days; from 5% to 95% of yearly growth completed)**													
**Control**	102	77	89	102	124	92	109	92	137	95	85	68	96
**Heated**	109	72	95	96	104	77	116	71	80	80	76	61	88
**Overall**	106	75	92	99	116	86	112	83	114	88	81	66	93
**Total annual growth (µm)**													
**Control**	1572	1672	1260	679	547	685	896	1292	1107	1055	1102	1065	1115
**Heated**	1018	1031	1079	824	818	1301	1513	1957	1409	1504	1547	1090	1216
**Overall**	1295	1322	1178	741	663	949	1160	1577	1228	1248	1293	1073	1160
**Average daily growth rate (µm/day)**													
**Control**	15.4	21.6	16.6	7.4	4.3	8.0	8.5	15.1	7.9	11.3	13.9	15.8	12.7
**Heated**	11.4	14.9	12.7	8.9	8.1	16.4	13.9	26.9	17.6	18.7	20.4	17.9	15.0
**Overall**	13.4	17.9	14.8	8.0	5.9	11.6	10.8	20.2	11.8	14.4	16.7	16.5	13.7

**Table 2 sensors-22-05155-t002:** Mixed between-subjects (control/heated) and within-subjects (years) analysis of variance of the trimmed means of parameters used to characterize the growth phenology of 7 balsam fir trees over the 2010–2021 period, excluding the 5 trees that did not survive the combined drought and defoliation events occurring in 2012.

Effect	Onset	Cessation			Duration			Total	Rate
	5%	80%	90%	95%	80%	90%	95%		
**Treatment**	0.2	2.0	5.0 *	9.6 *	2.8	5.6 *	11.2 **	8.4 **	21.0 ***
**Year**	85.1 ***	38.1 ***	14.4 ***	68.2 ***	3.5 *	3.0 *	19.0 ***	3.8 *	9.6 ***
**Interaction**	0.5	0.8	1.0	27.2 ***	1.6	1.0	2.9 *	0.9	1.3

Significance: * < 0.1, ** < 0.01, *** < 0.001.

**Table 3 sensors-22-05155-t003:** Spearman’s rank correlation coefficient among the parameters used to characterize growth phenology of 7 balsam fir trees over the 2010–2021 period, excluding the 5 trees that did not survive the combined drought and defoliation event occurring in 2012.

Effect		Cessation			Duration			Total	Rate
		80%	90%	95%	80%	90%	95%		
**Onset**	5%	0.51 ***	0.26 *	0.23 *	−0.28 **	−0.17	−0.05	−0.08	−0.04
**Cessation**	80%		0.49 ***	0.41 ***	0.59 ***	0.27 **	0.28 **	0.02	−0.10
	90%			0.68 ***	0.30 **	0.87 ***	0.60 ***	−0.24 *	−0.45 ***
	95%				0.22 *	0.55 ***	0.93 ***	−0.22 *	−0.59 ***
**Duration**	80%					0.46 ***	0.35 ***	0.11	−0.06
	90%						0.62 ***	−0.22 *	−0.44 ***
	95%							−0.21 *	−0.60 ***
**Total**									0.88 ***

Significance: * < 0.1, ** < 0.01, *** < 0.001.

## Data Availability

The data presented in this study are available on request from the corresponding author.

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
