# Peer review of "Characterizing Seasonal Radial Growth Dynamics of Balsam Fir in a Cold Environment Using Continuous Dendrometric Data: A Case Study in a 12-Year Soil Warming Experiment"

_sensors, 2022, doi:10.3390/s22145155_

Round 1
Reviewer 1 Report
Dear authors,
thanks, very interesting long time experiment. I have only two suggestions:
1. Please extend chapter 2.4. Statistical analyses. It is important for better understanding of your results
2. Extend Chapter 3.2. Correlations among the seasonal parameters and description of table in this chapter
Thanks
Author Response
Dear authors,
thanks, very interesting long-time experiment. I have only two suggestions:
Thank you very much.
- Please extend chapter 2.4. Statistical analyses. It is important for better understanding of your results
We have added more explanations about the statistical analyses performed. We have specified the parameters analyzed, the number of experimental units considered, as well as the type of relationship analyzed (monotonic) using the Spearman correlation test. - Extend Chapter 3.2. Correlations among the seasonal parameters and description of table in this chapter
We are not sure what additions the reviewer would like to this section (now section 3.3). We feel that this section is already well explained, i.e., the correlations among the seasonal parameters in relation to what is in table 3.
We are, however, open to an explicit suggestions.
Reviewer 2 Report
The importance of carrying out sequences of growth and development over long periods of time in the various ecosystems is and should be of interest to all. I congratulate the authors for their hard and consecutive work for over a decade, even when tree losses, were recorded during its study; however, it gives scientific evidence of the climatic changes that are occurring on the planet and the effects on the organisms that inhabit it. Excellent paper.

Author Response
Reviewer 2
The importance of carrying out sequences of growth and development over long periods of time in the various ecosystems is and should be of interest to all. I congratulate the authors for their hard and consecutive work for over a decade, even when tree losses, were recorded during its study; however, it gives scientific evidence of the climatic changes that are occurring on the planet and the effects on the organisms that inhabit it. Excellent paper.
Thank you very much.
We have made the suggested changes in the pdf file. We have moved the highlighted part to a new section in the results, i.e., to section 3.1 Soil temperature and dendrometer data.
Reviewer 3 Report
Dear Authors,
I think that the manuscript can be accepted for publication with only few minor changes.
The aim of the study is not clearly stated and this is a shortcoming of the Introduction.
I suggest underscoring the scientific value of the research in the conclusion section.
Author Response
Reviewer 3
Dear Authors,
I think that the manuscript can be accepted for publication with only few minor changes.
Thank you very much.
1. The aim of the study is not clearly stated and this is a shortcoming of the Introduction.
We are sorry for not being clear on the aims of the study. Although the aims were clearly stated, we understand that it might not have been apparent as we did not use the word “aims”. Now we have clearly stated the aims of the study in the last paragraph of the introduction.
2. I suggest underscoring the scientific value of the research in the conclusion section.
We did underscore the scientific value of the research, however, maybe we did not use the proper wording to do so. We have detailed the sentences indicating the scientific value of the research in the conclusion section.